# The Chagas non-endemic (ChaNoE) cohort: Aims and study protocol

**Pau Bosch-Nicolau**[1,2]*, **Juan María Herrero-Martínez**[3], **Marta Arsuaga**[2,4], **Sandra Chamorro-Tojeiro**[2,5], **Irene Carrillo**[6], **Carlos Bea-Serrano**[7], **Jara Llenas-García**[2,8,9,10], **Sandra Pérez-Recio**[11], **Elia Fernández-Pedregal**[12], **Clara Crespillo-Andújar**[2,5], **Aroa Silgado**[2,13], **Ana Pérez-Ayala**[14], **Fernando de la Calle-Prieto**[2,4], **Laura Prieto-Pérez**[6], **José A. Pérez-Molina**[2,5], **Israel Molina**[1,2]

**1** International Health Unit Vall d'Hebron-Drassanes, Infectious Diseases Department, Vall d'Hebron University Hospital, PROSICS Barcelona, Barcelona, Spain, **2** Centro de Investigación Biomédica en Red de Enfermedades Infecciosas (CIBERINFEC), Instituto de Salud Carlos III, Madrid, Spain, **3** Internal Medicine Department, University Hospital 12 de Octubre, Madrid, Spain, **4** National Referral Centre for Tropical Diseases and International Health, High Level Isolation Unit, La Paz-Carlos III University Hospital, Madrid, Spain, **5** National Referral Centre for Tropical Diseases, Infectious Diseases Department, Ramón y Cajal University Hospital, IRYCIS, Madrid, Spain, **6** Internal Medicine-Infectious Diseases Department, University Hospital Fundación Jiménez Díaz, Madrid, Spain, **7** Infectious Disease Unit, Internal Medicine Department, Clinic University Hospital of Valencia, Valencia, Spain, **8** Internal Medicine Service Vega Baja Hospital, Orihuela, Spain, **9** Foundation for the Promotion of Health and Biomedical Research of the Valencian Community (FISABIO), Valencia, Spain **10** Clinical Medicine Department, Miguel Hernández University, Elche, Spain, **11** Department of Infectious Diseases, Bellvitge University Hospital Bellvitge University Hospital-Bellvitge Biomedical Research Institute (IDIBELL), University of Barcelona, L'Hospitalet de Llobregat, Spain, **12** International Health Program (PROSICS), Direcció Territorial de Malalties Infeccioses Metropolitana Nord, Fundació Lluita contra les Infeccions, Infectious Diseases Department, Hospital Universitari Germans Trias i Pujol, Badalona, Spain, **13** Microbiology Department, Vall d'Hebron University Hospital, PROSICS Barcelona, Barcelona, Spain, **14** Microbiology Department, University Hospital 12 de Octubre, Madrid, Spain

* pau.bosch@vallhebron.cat

## Abstract

Chagas disease (CD), caused by *Trypanosoma cruzi*, is a neglected tropical disease with significant public health implications. While its primary transmission occurs in endemic regions via triatomine vectors, migratory processes have led to an increased prevalence in non-endemic areas as in Spain where an estimated 50,000 people live with CD. Chronic Chagas cardiomyopathy (CCC) and digestive complications are the primary manifestations, yet diagnostic criteria, especially regarding organic involvement, and treatment indications are still a matter of debate. There is an urgent need for standardized approaches to improve care and identify biomarkers for progression and treatment response. The Chagas non-endemic cohort (ChaNoE) aims to recruit individuals with chronic CD across multiple centers in Spain. Inclusion criteria involve a confirmed diagnosis based on two serological tests. Participants will receive comprehensive diagnostic evaluations, including electrocardiography, echocardiography, and periodic serological and PCR assessments. Follow-up will focus on disease progression, particularly CCC and digestive involvement, using standardized protocols. The study also establishes a biobank for serum samples to facilitate biomarker research. The ChaNoE cohort addresses critical gaps in the understanding of CD in non-endemic regions. By standardizing diagnostic and treatment

**Data availability statement:** No datasets were generated or analysed during the current study. All relevant data from this study will be made available upon study completion.

**Funding:** This project was funded by "Proyectos de investigación en salud" program of the Instituto Carlos III, Ministry of Science, Innovation and Universities of the Spanish Government. Grant number: PI22/01894". The funders had no role in study design, data collection and analysis, decision to publish, or preparation of the manuscript.

**Competing interests:** The authors have declared that no competing interests exist.

protocols, it seeks to harmonize care and enable comparisons with cohorts in endemic areas. The creation of a biobank supports the identification of biomarkers for disease progression and treatment efficacy, a current unmet need in CD management. This initiative also strengthens research networks and informs public health strategies to mitigate the burden of CD in non-endemic settings. Findings will be disseminated to key stakeholders to improve the clinical and epidemiological understanding of this neglected disease.

## Introduction

Chagas disease (CD), or American trypanosomiasis, caused by the protozoan *Trypanosoma cruzi* is one of the most significant neglected tropical diseases due to its high prevalence and the limited attention devoted to research. Although the overall incidence has declined owing to improvements in vector control, disease detection, and treatment, its introduction in Europe has notably increased in recent decades due to migratory processes. It is estimated that more than 50.000 people with CD reside in Spain, with an underdiagnosis rate of 71% and an overall undertreatment ratio of 82.5%, representing a major public health challenge [1,2]. In endemic regions, the primary transmission route of CD is vectorial, through triatomine insects. However, in non-endemic regions, mother-to-child transmission and infections through blood or organ donations can occur [3]. To mitigate this risk, health authorities have introduced various screening programs for blood and organ donations along with perinatal and prenatal screening protocols but there is still much heterogeneity in its coverage and implementation [4].

Following an acute phase, often asymptomatic, infected individuals enter into an indeterminate chronic phase characterized by the absence of symptoms and the normal results of complementary examinations. However, approximately 30% of individuals will develop cardiac or digestive manifestations with a latency period that can last 10–30 years. Chronic chagasic cardiomyopathy (CCC) is the most severe manifestation leading to electrocardiographic disturbances and motility alterations that can progress to heart failure, malignant arrhythmias, and thromboembolic events [5]. Regardless of its potential severity, there remains a lack of validated markers to predict the onset and progression of visceral involvement. Currently, treatment is recommended for people in the chronic phase of CD before moderate to severe cardiomyopathy is established, despite there is limited evidence on efficacy and the toxic profile of available medications. This indication is based on the improved prognosis for treated patients regarding CCC development and the prevention of vertical transmission when treating women of childbearing age [6–8]. However, the only accepted cure criterion is serological negativization, which can occur more than 10–20 years after treatment, leading to lifelong follow-up of these individuals [9].

Our country represents a model for non-endemic countries considering the implementation of health policies to diagnose and treat the disease. Even if cohorts of individuals living with CD already exist, the follow-up and evaluation criteria are heterogeneous, hindering the acquisition of consistent and comparable data. The aim of the present study is to establish a multicenter cohort that standardizes and validates follow-up and treatment criteria. This approach would facilitate an accurate assessment of the disease and its complications, improving the quality of clinical management and implementing more effective and uniform therapeutic and follow-up strategies ultimately enhancing the quality of life of affected individuals.

## Methods

### Aims and objectives

The Chagas non-endemic cohort (ChaNoE) cohort aims to establish a cohort of individuals with CD living in non-endemic areas, ensuring they receive the accepted standard of care

for the disease and offer them a prospective follow-up throughout their lives. Setting up this cohort will facilitate the homogenization of disease management and follow-up in non-endemic settings, providing current evidence-based strategies for affected individuals and protocol guidance for less familiarized caregivers. Additionally, this cohort will facilitate comparisons with other cohorts, particularly those in endemic countries, allowing us to track the condition from early factors to long-term impacts. With the creation of a serum bank, we will also be able to investigate new and emerging biomarkers in CD, addressing one of the current critical issues in the field.

The overall study objectives are to:

1. Describe the epidemiology and sociodemographic characteristics of individuals living with CD in a non-endemic region.

2. Evaluate the evolution of the disease with a special focus on identifying the risk factors associated with its progression.

3. Assess treatment efficacy and tolerance.

4. Establish a prospective serum bank to explore and validate future biomarkers for cure and progression.

5. Stimulate clinical, epidemiological and biological research on CD within the participating centers.

Primary outcomes:

- All-cause mortality

- Heart transplant requirement

- Cure, defined as the negativization of *T. cruzi* antibodies in accordance with current recommendations [5].

Secondary outcomes:

- Hospitalization due to heart failure or arrythmia

- Cardiac device implantation

- Neuro-vascular disease

- Disease progression, including cardiac and/or digestive forms. Cardiac progression will be defined as any transition from the indeterminate to cardiac stage or any progression within the proposed classification stages.

- Detection of *T. cruzi* qPCR positivity in peripheral blood following after complete treatment.

## The "ChaNoE" cohort

The cohort will prospectively include people living with chronic CD who are attending any of the participating centers. In Spain, universal health coverage ensures that patients are typically referred to specialized centers from primary care, blood banks or targeted community campaigns. At this point, the current participating centers include: University Hospital of Bellvitge (Hospitalet de Llobregat), Clinic University Hospital of Valencia (Valencia), University Hospital Doce de Octubre (Madrid), University Hospital Germans Trias i Pujol (Badalona), Jiménez Díaz Foundation (Madrid), University Hospital La Paz (Madrid), University Hospital Ramón

y Cajal (Madrid), University Hospital Vall d'Hebron (Barcelona) and Hospital Vega Baja (Orihuela). Participation remains open to additional centers. These institutions already provide comprehensive, multidisciplinary care for individuals living with CD, involving specialists in infectious diseases, internal medicine, microbiology, cardiology and gastroenterology.

Inclusion criteria:

- Confirmed chronic Chagas disease through *T. cruzi* detection on two distinct serological tests

- Signed informed consent

Exclusion criteria:

- Acute CD cases.

- Congenital infections receiving early treatment and achieving serological negativization

- Pregnant individuals

Participant recruitment commenced on August 1, 2023, and the first analysis is planned for August 1, 2026. The study is designed as an open-ended cohort with no predefined conclusion date. To minimize loss of follow-up, patients who miss scheduled visits will receive reminders via email or telephone. Fig 1 shows the general flowchart of individuals included in the cohort.

## Laboratory diagnosis and follow-up of patients with chronic *T. cruzi* infection

Due to the lack of a serological reference test that alone achieves 100% sensitivity and specificity, CD diagnosis will be based on two serological tests using different principles and antigens [10]. Following the latest Spanish consensus, the recommended algorithm for suspected chronic *T. cruzi* infection consists of the use of an initial high-sensitivity assay, such as chemiluminescence immunoassays (CLIA) or chemiluminescence microparticle immunoassay

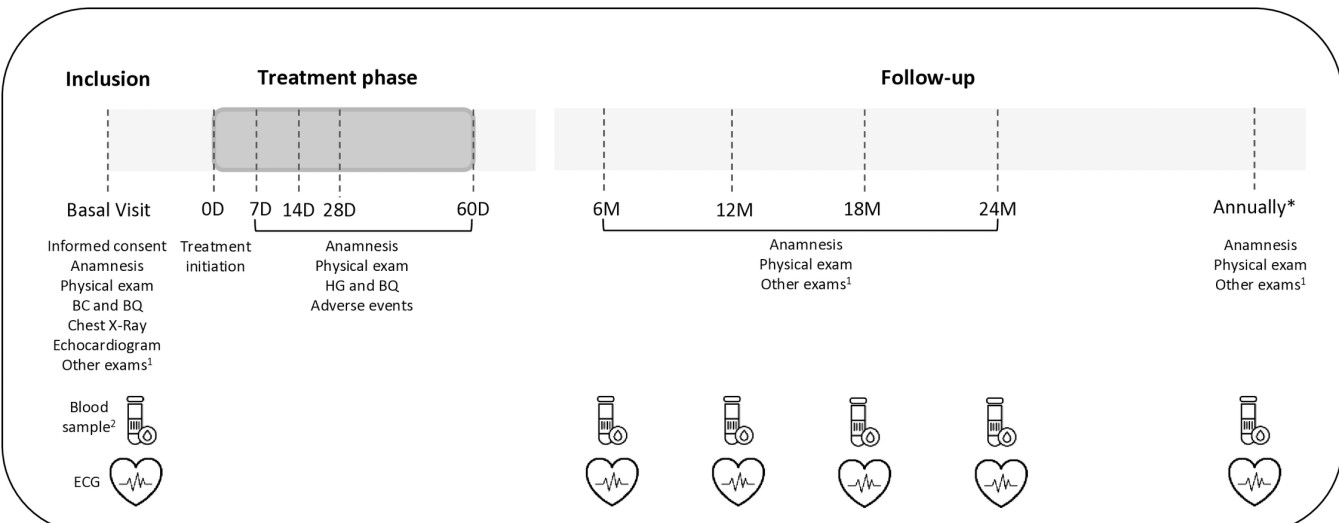

**Fig 1. General flowchart.** Abbreviations: BC, blood count; BQ, biochemistry; D, days; ECG, electrocardiogram; M, months. [1] Other test: cardiological or digestive exams depending on symptoms and the grade of organic involvement as mentioned in methods section. [2] Blood samples: qPCR, serology and the biobank serum sample.

(CMIA), followed by confirmation with a high specificity assay, such as enzyme-linked immu-nosorbent assays (ELISA) or rapid diagnostic test (RDT) [11].

Regarding PCR, each center will use its technique upon availability, though a protocol targeting satellite DNA is recommended as among the various protocols and commercial kits available, those targeting satellite DNA have shown better performance [12].

Serological and PCR assessments will be conducted every 6 months during the first two years of follow-up and then annually. A positive PCR result at any follow-up visit after treatment completion will be considered a parasitological failure, and the participants will be offered the alternative treatment.

## Chagasic cardiomyopathy: definition, classification and diagnostic approach

Chagas cardiomyopathy (CCC) is the most common visceral manifestation with an estimated annual incidence of 1.85 to 7% [13–15]. In our cohort, its diagnosis will be based on electro-cardiographic findings and the following cardiac imaging techniques.

Table 1 presents defining ECG findings of CCC in our cohort, considering the limitations of the different existing definitions and that none of these findings are pathognomonic.

Echocardiography will also be used for CCC assessment. It will be defined by the presence of segmental or global left ventricle (LV) or right ventricle (RV) systolic function impair-ment, increased end-diastolic diameter of the LV (>55mm), ventricular aneurysms or cardiac thrombus even in the presence of a normal ECG [18]. We will also record other common echocardiographic findings as diastolic dysfunction, additional cavity dilatations, or secondary valve alterations. We will use the reference normal values from the American Society of Echo-cardiography and the European Society of Cardiovascular Imaging [19]. As an exception, the guideline from the Brazilian Society of Cardiology will be followed, considering a LV ejection fraction (LVEF) ≥55% using 2D volumetric data as normal [16].

Other potential causes of heart disease that could explain abnormalities (such as ischemic heart disease or other causes of dilated cardiomyopathy) should likewise be excluded. Addi-tional tests will be performed considering their availability and other patient's risk factors such as cardiac MRI, CT scan, myocardial stress test, or coronary angiography. Furthermore, the evolution over time will be monitored to establish a possible causal relationship with CD.

Table 1. Electrocardiographic alterations in Chagas disease.

| Typical Changes (Defining of CCC) | Nonspecific Changes (Non-defining) |
|---|---|
| • Sinus bradycardia < 40 bpm<br>• Sinus node dysfunction<br>• Second-degree or third-degree atrioventricular block<br>• Complete RBBB<br>• RBBB with left anterior fascicular block<br>• Atrial fibrillation or atrial flutter<br>• Frequent premature ventricular complexes (>1 per ECG, often polymorphic)<br>• Ventricular tachycardia (sustained and non-sustained)<br>• Ventricular fibrillation<br>• Primary ST segment and T wave abnormalities<br>• Pathological Q waves or electric inactive areas | • Sinus bradycardia ≥ 40 bpm<br>• First-degree atrioventricular block<br>• Incomplete right or left bundle branch block<br>• Isolated left anterior fascicular block<br>• Mean QRS electrical axis deviation to the left<br>• Sinus arrhythmia and sinus tachycardia<br>• Isolated premature ventricular beats<br>• Low voltage in limb leads (<0.5 mV)<br>• Non-specific ST segment and T wave changes<br>• Wandering atrial pacemaker |

(Modified from Marín-Neto JA *et al.* [16] and Hasslocher-Moreno AM *et al.* [17]).

Abbreviations: AV, atrioventricular; AV block, atrioventricular block; bpm, beats per minute; CCC, chronic Chagas cardiomyopathy; ECG, electrocardiogram; RBBB, right bundle branch block.

Table 2 summarizes the proposed classification and follow-up based on the Brazilian Society of Cardiology [16]. Asymptomatic patients with a normal ECG are classified as stage A or indeterminate form. The presence of typical electrocardiographic abnormalities defines the stage change, assuming cardiac involvement. These stages are defined as follows: B1, if there are no echocardiographic abnormalities (or other imaging tests), or if abnormalities are present, the LVEF is normal; B2, if there is left ventricular systolic dysfunction with LVEF <55% in the absence of heart failure symptoms; C, patients with ventricular dysfunction and current or previous symptoms of heart failure; D, patients with refractory symptoms of heart failure at rest (NYHA class IV) despite optimized treatment. Patients with electrocardiographic abnormalities not definitive of CCC (non-specific, see Table 1) will be classified as stage A with non-specific electrocardiographic changes, unless additional studies (such as echocardiography, Holter monitoring, cardiac MRI, or others) reveal CCC abnormalities of other stages in the absence of other causes of heart disease.

The initial evaluation will include a focused medical history and physical examination, complete blood count and biochemistry, chest X-ray, 12-lead ECG with a rhythm strip of at least 30 seconds or 24h-Holter monitoring, a transthoracic echocardiogram and NYHA class classification. For patients with symptoms or ECG findings suggestive of cardiac involvement, ambulatory 24-hour ambulatory ECG monitoring will be performed, with additional tests based on clinical presentation and differential diagnosis. During follow-up, an annual ECG and reassessment of symptoms, cardiac signs, and NYHA classification are recommended, as asymptomatic patients with a normal ECG have a favorable prognosis if they remain in this condition [16]. Echocardiographic follow-up will be guided primarily by the latest World Heart Federation consensus recommendations [20]. This periodicity will be more frequent in advanced stages based on current knowledge of the natural history of the disease [14]. Depending on the progression, additional cardiac examinations may be required, such as 3D echocardiography, cardiac MRI, prolonged ambulatory electrocardiographic monitoring using an external recorder or implantable Holter monitor, electrophysiological studies, stress test, catheterization, or other nuclear medicine tests [14,16,21].

## Digestive involvement: definition, classification and diagnostic approach

In our setting, digestive tract involvement in CD affects up to 15–20% of individuals with chronic CD, and it seems to be more common in patients coming from the Southern cone [5,22,23]. The clinical presentation ranges from asymptomatic motility disorders to progression to severe forms with megaviscera (especially in the colon and esophagus). Table 3 summarizes the signs and symptoms of digestive involvement that will be recorded in the present cohort.

Specific follow-up with complementary explorations for individuals with digestive symptoms will depend on clinical progression and evolution. A thorough physical examination and detailed medical history, including dietary habits and symptoms, will be obtained at baseline and during each visit. Tests to assess digestive involvement will be recommended if clinical symptoms are present at baseline or during follow-up. Although up to 11% of asymptomatic individuals can exhibit digestive alterations, follow-up tests are not standardized [23].

For initial evaluation of digestive involvement, esophagogram and barium enema will be the choice tests. Esophageal manometry will be recommended for individuals with esophageal symptoms that affect their quality of life, even if the esophagogram is normal. Endoscopic studies will be reserved for cases where megaesophagus is present to assess the status of the esophageal mucosa or when underlying esophageal/colonic disease is suspected [23]. Individuals with no specific findings in the complementary tests will be evaluated to rule out other

**Table 2. Classification and follow-up of adult individuals with chronic CD regarding cardiac disease[1].**

| Clinical form | | Indeterminate | Chronic Chagasic Cardiomyopathy | | | |
|---|---|---|---|---|---|---|
| **Stages AHA/ACC** | | **A** | **B1** | **B2** | **C** | **D** |
| **Definition** | | **No evidence of structural cardiopathy** | **Structural cardiopathy** | | | |
| | | | **Normal LVEF** | **Reduced global systolic function** | | |
| | | Asymptomatic | Asymptomatic | Asymptomatic | Heart failure (current or previous) | Heart failure (refractory symptoms at rest) |
| **Characteristics** | ECG | Normal | Abnormal[2] | Abnormal | Abnormal | Abnormal |
| | Segmental wall motion abnormalities | Usually absent | Can be present | Can be present | Can be present | Can be present |
| | LVEF (Echo-Simpson) | ≥ 55% | ≥ 55% | **< 55%** (usually between 41–54%) | < 55% (usually ≤ 40%) | Usually ≤ 25% |
| | NYHA FC | N/A | I | I | I-IV | IV |
| | Cardiomegaly (chest X-ray) | Absent | Absent | Can be present | Usually present | Present |
| | Complex ventricular arrhythmia (24h-Holter) | Usually absent | Can be present | Usually present | Present | Present |
| | Myocardial fibrosis (MRI) | Can be present | Usually present | Usually present | Present | Present |
| **Basal explorations** | | | | | | |
| **Anamnesis/Physical exam** | | X | X | X | X | X |
| **Chest x-Ray** | | X | X | X | X | X |
| **ECG[3]** | | X | X | X | X | X |
| **Echocardiography** | | X | x | X | X | X |
| **24h-Holter** | | If LV segmental wall motion abnormalities<br>If symptoms (palpitations, dizziness, pre-syncope or syncope) or arrhythmia (frequent premature ventricular beats) | | | | |
| | | | X | X | X | X |
| **LABORATORY (peripheral blood)** | | | | | | |
| **Hemogram and biochemistry** | | X | X | X | X | X |
| **Serology** | | X | X | X | X | X |
| **PCR** | | X | X | X | X | X |
| **Biobank** | | X | X | X | X | X |
| **Follow-up** | | | | | | |
| **Anamnesis** | | X | X | X | X | X |
| **Chest X-ray or lung ultrasound** | | If heart failure or dyspnea | | | | |
| **ECG[3]** | | ANNUAL[6]: normal ECG or minor changes | | Every 6 MONTHS | Every 6 MONTHS | Every 6 MONTHS |
| | | | Every 6 MONTHS if major changes[4]. | | | |
| | | New ECG if there is a clinical change | | | | |
| **Echocardiography** | | Every 3–5 YEARS | Every 2–3 YEARS | Every 1–2 YEARS[5] | | Individualized[5] |
| | | If worsening of heart failure, embolic event, or changes in the ECG | | | | |
| **Holter 24 h** | | If there are LV segmental wall motion abnormalities<br>If there are symptoms (syncope or pre-syncope) or frequent arrhythmias on physical examination.<br>If there are major changes on the ECG[3] | | | | |

*(Continued)*

**Table 2.** (Continued)

| Clinical form | Indeterminate | Chronic Chagasic Cardiomyopathy | | | |
|---|---|---|---|---|---|
| LABORATORY (peripheral blood) | | | | | |
| Blood count and biochemistry | ANNUAL[6] | ANNUAL[6] | ANNUAL[6] | ANNUAL[6] | ANNUAL[6] |
| Serology | | | | | |
| PCR | | | | | |
| Biobank | | | | | |

(Modified from Nunes MCP *et al.* [14], Marín-Neto JA *et al.* [16] and Ralston K *et al.* [20]).

Abbreviations: AHA/ACC (American Heart Association and American College of Cardiology); ECG: electrocardiogram; FC: functional class; LVEF: left ventricle ejection fraction; MRI: magnetic resonance imaging; N/A: Non-applicable; NYHA: New York Heart Association; PCR: polymerase chain reaction.

[1] In adult immunocompetent and non-pregnant individuals with chronic CD.

[2] Patients with a normal ECG but echocardiographic alterations will also be included in this classification.

[3] 12-lead electrocardiogram and 30-second rhythm strip in lead II.

[4] Major ECG changes: sinus node dysfunction, atrioventricular block or frequent premature ventricular complexes.

[5] At 3–6 months when in case of optimization of heart failure treatment with reduced ejection fraction (HFrEF), to assess the need for additional pharmacological treatments and/or advanced therapies.

[6] During the first two years, ECG and laboratory determinations will be performed every 6 months in all patients.

**Table 3.** Signs and symptoms of digestive involvement.

| Localization of digestive involvement | Signs and Symptoms |
|---|---|
| Esophageal alterations | • Dysphagia<br>• Regurgitation (generally liquids with food remnants)<br>• Retrosternal chest pain<br>• Odynophagia<br>• Nocturnal cough<br>• Sialorrhoea<br>• Parotid gland hypertrophy |
| Gastric/ duodenal alterations | • Dyspepsia<br>• Pyrosis<br>• Bloating<br>• Satiety sensation<br>• Epigastric pain |
| Colonic alterations | • Constipation<br>• Diarrhea<br>• Changes in bowel habits<br>• Straining during bowel movements<br>• Rectal tenesmus |

*Adapted from Pinazo *et al.* [23].

possible underlying diseases, and symptomatic treatment will be offered. Tables 4 and 5 summarize the classification of esophageal and colon disease that we will use.

Specific treatment for digestive involvement depends on the patient's symptoms. Hygienic-dietary measures and laxatives are usually sufficient for colonic involvement. In more advanced cases, surgical intervention will be considered. No specific treatments can restore esophageal involvement completely, but partial recovery is possible through medical, endoscopic and surgical interventions [23]. Long-term follow-up of these individuals will be coordinated with the specialists involved.

**Table 4. Classification of chagasic esophageal disease.**

| Classification | Findings |
|---|---|
| Group 0 | Asymptomatic + without radiological findings |
| Group I | Apparently normal caliber of esophagus.<br>Slow transit with minor retention<br>Air in gastric fundus |
| Group II | Small-moderate increase in caliber<br>Appreciable contrast retention<br>Frequent tertiary waves +/− hypotonus<br>Air in gastric fundus |
| Group III | Large increase in diameter<br>Important contrast retention<br>Reduced motor activity + hypotonus<br>No air in gastric fundus |
| Group IV | Dolichomegaesophagus<br>Great retention<br>Atonic<br>No air in gastric fundus |

*Adapted from Rezende *et al.* [24].

**Table 5. Classification of chagasic colon disease.**

| Classification | Findings |
|---|---|
| Group 0 | No alterations in barium enema |
| Group 1 | Patients with dolichocolon/ dolichosigma |
| Group 2 | Dolichomegacolon• Descending colon >6.5 cm diameter<br>• Ascending colon >8 cm in diameter<br>• Caecum >12 cm in diameter |

*Adapted from Pinazo *et al.* [23].

**Table 6. Recommended doses of Benznidazole and Nifurtimox.**

| Age group and Weight | Benznidazole | Nifurtimox |
|---|---|---|
| Children < 12 years or <40 Kg | 10 mg/Kg/day divided bid | 10-20 mg/Kg/day |
| Children ≥ 12 years or ≥40 Kg | 5–7 mg/Kg/day bid | 8-10 mg/Kg/day |
| Adult <50 Kg | 5 mg/Kg/day (bid or tid) | 120 mg tid |
| Adult 50–70 Kg | 5 mg/Kg/day (bid or tid) | 180 mg tid |
| Adult ≥ 70 Kg[1] | 5 mg/Kg/day (bid or tid) | 240 mg tid |

* ChaNoE consensus modified from *Pérez-Molina et. al* [27].

[1]Maximum dose 400mg/day. Treatment extension can be individualized in extreme weights.

## Treatment: therapeutic schemes and follow-up

Two drugs, benznidazole and nifurtimox, are available for treating CD. Both drugs offer similar efficacy and effectiveness, so the choice between them is usually based on availability [25]. In our cohort, the recommended first-line treatment is benznidazole, but nifurtimox will be also an option. Regimens for immunocompetent individuals are:

- Benznidazole: with the dose adjusted on body weight to 5–10mg/kg/day, divided into 2–3 doses, for 60 days.

- Nifurtimox: with the dose similarly adjusted to 8–10 mg/kg/day, for a duration of 60–90 days.

Table 6 summarizes recommended doses of each drug. The effectiveness of these drugs varies depending on the stage of the disease. In the acute phase or in cases of congenital Chagas, treatment efficacy is around 80–100%. However, in the indeterminate chronic phase, the cure rate decreases to 20–40% [26]. Table 7 shows proposed recommendations for antiparasitic treatment of CD, according to the level of evidence.

**Table 7. Recommendations for antiparasitic treatment.**

| Chagas disease treatment | Strength and quality of the recommendation |
|---|---|
| *Always recommended* | |
| Acute infection | AI |
| Congenital infection | AI |
| Children ≤12 years with chronic disease | AI |
| Children 13–18 years with chronic infection | AIII |
| Reactivation of T. *cruzi* in people with HIV or immunosuppressed patients | AII |
| *Generally recommended* | |
| Women of child-bearing age | BII |
| Adults in the indeterminate form or mild cardiomyopathy | BII |
| Patients undergoing immunosuppression therapy | BII |
| *Optional treatment* | |
| Patients with digestive form without severe cardiomyopathy | CII |
| *Generally not recommended* | |
| Patients with moderate to severe cardiomyopathy[1] | DI |
| Megaesophagus[2] | DIII |
| Lactation[3] | DIII |
| Psychiatric disorders (specially Nifurtimox) | DIII |
| *Never recommended* | |
| During pregnancy | EIII |
| Acute severe renal or liver impairment (relative contraindication) | EIII |

\* ChaNoE consensus.

**A:** very solid evidence to support the use of a recommendation; it would always be provided. **B:** moderate evidence to support the use of a recommendation; it would generally be offered. **C:** limited evidence to support a recommendation; optional. **D:** moderate evidence against a recommendation; it would generally not be offered. **E:** very solid evidence against a recommendation; it would never be offered. **I:** evidence obtained from at least one randomized clinical trial. **II:** evidence obtained from at least one well-designed non-randomized trial, cohort studies, case-control analytic studies (preferably from more than one center), time series, or dramatic results in uncontrolled experiments. **III:** evidence from expert opinions based on clinical experience or descriptive studies.

[1]Stage B1 and B2 with severe arrhythmias, stages C and D.

[2]Dilation, surgical correction or rehabilitation of megaesophagus is recommended before initiating the treatment to ensure drug transit and absorption.

[3]Treatment during lactation is generally not recommended although transference of benznidazole and nifurtimox into breast milk is unlikely to present a risk [28,29]. It could be individualized in the acute phase, immunosuppressed women or when avoiding these medications could cause a prolonged delay of treatment.

Close monitoring during treatment is essential. Patients will have direct access to their healthcare provider through a designated pathway (emergency department or unplanned visits) to ensure prompt evaluation if any side effects arise, allowing for quick initiation of symptomatic treatment and temporary or definitive discontinuation of the medication. The most common adverse events and the suggested management strategies are outlined in Table 8. Each symptom should be carefully assessed. Each physician will decide whether to temporarily interrupt treatment, introduce symptomatic treatments (such as steroids, antihistamines or acid suppressant drugs), or permanently discontinue the medication. We recommend conducting one or two blood tests during treatment, including a blood count and biochemistry tests assessing renal and hepatic function. We recommend performing these tests 15 days after starting treatment and again during the fourth to fifth week of treatment [30].

## Biomarkers: biobank creation

Besides creating a prospective clinical cohort of individuals with CD using uniform criteria, one of our main objectives is to generate a comprehensive serum library. This biobank has already been registered in the National Biobank Registry (Collection C.0004055) and is aimed at facilitating the study and validation of future biomarkers for CD.

Sample collection for biomarker determination will be carried out at the baseline visit and at each follow-up visit once the treatment is completed. Samples will consist of aliquoted serum samples stored at −80ºC to ensure long-term viability and integrity. Each center will be responsible for maintaining the samples until their analysis is decided.

**Table 8. Adverse events, time of onset and management [31,32].**

| SIDE EFEFCT | TIME OF ONSET AND TREATMENT |
|---|---|
| **Benznidazole** | |
| **Skin**: disseminated macules, skin itching, urticaria, DRESS, Stevens-Johnson syndrome | Onset: around day 10–15 of treatment. Treatment: antihistamines, corticosteroids, and/or discontinuation of benznidazole. |
| **Digestive**: nausea, vomiting, abdominal pain, jaundice, hepatomegaly | Onset: during the first two weeks. Treatment: symptomatic treatment and emphasize taking the medication with meals; adjust diet. |
| **Neurological**: headache, peripheral polyneuropathy, sensory disturbances | Onset: around day 40 of treatment. Treatment: symptomatic. Sometimes, it is necessary to discontinue the medication |
| **Bone Marrow Suppression**: neutropenia, anemia, agranulocytosis with fever | Onset: around 2–3 weeks of treatment. Treatment: drug interruption |
| **Nifurtimox** | |
| **Digestive**: anorexia, nausea, weight loss | Onset: first 3 weeks of treatment and may persist throughout the treatment. Treatment: symptomatic treatment and emphasize taking the medication with meals |
| **Psychiatric**: depression, psychomotor agitation | Onset: appears in the first 2–3 weeks. Treatment: it may be necessary to discontinue the medication. |
| **Neurological**: headache, dizziness, peripheral polyneuropathy, paresthesia in palms and soles (dose-dependent, resolves over months) | Onset: between weeks 2 and 10 of treatment. Treatment: symptomatic. Sometimes it is necessary to discontinue the medication. |

Treatment for immunosuppressed individuals would be individualized on a case-by-case basis, with treatment regimens and secondary prophylaxis tailored to the immunosuppressive condition and type of reactivation [5]. Similarly, microbiological monitoring will be adjusted for each individual case.

Initially, these samples will be used to test the proposed metabolites as potential biomarkers for disease progression and cure. However, the biobank is designed as a long-term resource to facilitate future research, allowing for the evaluation of new molecular targets as they emerge. Access to the biobank will be regulated according to current legislation on biological samples, ensuring compliance with ethical and legal frameworks. Future research projects, whether internal or from third parties, may request access to the samples, provided they align with the study's objectives and adhere to all ethical and legal requirements. Decisions regarding sample use will be overseen by a designated biobank committee, which will assess proposals based on scientific merit, feasibility, and compliance with applicable regulations.

## Data management and statistical analysis

Participant data will be de-identified, as each individual will be assigned a unique and linked study ID number. The principal investigator of each center will be the only responsible for the linkage between patient data and the study ID number in their center. Individuals' unique study ID numbers will be used at all times, within data storage and data analysis, to ensure anonymity and confidentiality throughout all stages of the study. Participants will be informed that they can discontinue the study at any time.

The study's database will be centralized and accessible online, with individualized access keys for each participating site. Data entry personnel at each center will only have access to their own data, with permissions limited to entering new records. Each site will have a designated data monitor responsible for overseeing and managing data, with access restricted to their own center's records. The central team will have viewing privileges and editing rights only in cases where inconsistencies are identified by local teams. Data quality will be monitored. Checks will include missing data, numerical and date range validation, and overall data consistency.

The statistical analysis will include a detailed descriptive analysis. Continuous variables will be summarized using means and standard deviations for normally distributed data, and medians and interquartile ranges for non-normally distributed data. Categorical variables will be presented as frequencies and percentages. We will employ the student's t-test for both paired and independent samples to compare continuous variables with a normal distribution. For non-normally distributed continuous variables, the Mann-Whitney U test will be used. Categorical data comparisons will utilize the McNemar test. ANOVA will be applied to assess differences in biomarkers between groups. A bidirectional ANOVA with repeated measures will also be used to identify metabolites that exhibit significant effects across the main experimental groups.

Survival outcomes will be analyzed using Kaplan-Meier curves, and differences between groups will be evaluated with the log-rank test. Cox proportional hazards regression models will be used to estimate hazard ratios (HRs) and 95% confidence intervals (CIs) for the association between risk factors, treatment, and biomarkers with survival outcomes. The proportional hazards assumption will be assessed using Schoenfeld residuals. Multivariable models will be constructed to adjust for potential confounders, including relevant demographic, clinical, and laboratory variables. Model selection will be based on clinical relevance and statistical criteria. Interaction terms and potential effect modification will also be explored. Additional exploratory analyses may be conducted, including time-dependent covariates or alternative modeling approaches, depending on data distribution and emerging findings. All significance tests will be two-sided, with an alpha level of 0.05 unless otherwise specified.

### Ethical considerations and dissemination plans

The study protocol has been approved by the institutional review board of the coordinating center at Vall d'Hebron University Hospital in Barcelona (ID number: EOM(AG)028/2023(6141)), as well as from the ethics committee of each participating site. All participants will provide written informed consent prior to enrollment in the study. For minors, informed consent will be obtained from their parents or legal guardians, in accordance with ethical and legal requirements.

The dataset including individual participant data and a data dictionary defining each field in the set will be available with publication to appropriate academic parties on request to the chief investigator (israel.molina@vallhebron.cat) in accordance with the data sharing policies of Vall Hebron Research Institute, with input from the co-investigator group where applicable.

The results will be disseminated to patient organizations, key stakeholders and governments. Additionally, findings will be presented at scientific meetings and conferences and published in high-impact journals. This data will help researchers, governing bodies, and patients themselves, gain a more accurate understanding of some of the health challenges they experience throughout their lives.

## Discussion

Our knowledge of the clinical manifestations and natural evolution of CD is still scarce. Much of the data we rely on comes from studies conducted over 50 years ago in rural areas of endemic regions, where the social and economic context were quite different from those of today. This fact presents a significant challenge when trying to extrapolate this information to our current understanding of CD. Moreover, both the parasitological tests used for diagnosing the disease and the techniques for evaluating its progression have evolved considerably since then [33]. In addition, other epidemiological factors, such as the risk of reinfection, hygienic-dietary improvements or simply access to health systems have improved significantly in recent decades.

These different temporal scenarios become even more complex when considering regional differences. Growing evidence highlights the relevance of the parasite's genetic diversity in shaping the disease's clinical spectrum and its response to treatment [34,35]. Additionally, data from series of chronic CD in non-endemic countries reveal a different clinical profile, characterized by younger individuals and lower morbidity. While cardiac involvement occurs in 14–45% of chronically infected individuals and gastrointestinal involvement in 15–20% of chronic cases, a recent systematic review reported higher prevalence of Chagas cardiomyopathy [42.7% (37.3–48.4)], while the digestive form remained similar [13.3% (9.1–19.0)] [5,36]. Interestingly, clinical manifestations are more frequent in endemic areas compared to non-endemic settings: cardiomyopathy 48% vs. 23.1% and digestive involvement (21.7% vs 8.5%). Moreover, the slow progression of cardiac disease (annual incidence of 1.85–7% in untreated individuals) necessitates long-term follow-up of large cohorts to accurately define its natural course [15]. This need for extended observation and larger sample sizes becomes even more marked when considering the effect of antiparasitic treatment.

Another significant limitation is the absence of reliable cure criterion. Currently, the only accepted cure criterion is seronegativization, a process that may take 10–20 years after treatment. PCR-based assays have been explored as surrogate markers but exhibit low sensitivity due to intermittent low-level parasitemia characteristic of this phase [12]. However, PCR would be useful for detecting parasite load after anti-parasitic treatment, acting as a marker for parasitological failure [37,38]. The absence of a cure biomarker that could assess treatment efficacy within a reasonable timeframe also makes it difficult the design of new clinical trials

for evaluating new drugs, making its identification a top research priority. To achieve this, one of the most widely used strategies is the analysis of well-categorized cohorts with clearly defined clinical outcomes. Although potential serological, biochemical and hypercoagulability biomarkers have been proposed, none have been validated to date [39,40]. Preliminary metabolomic profiles suggest promising candidates for disease progression and cure markers [41]. The biobank created through this study will contribute to validating these biomarkers and support future research.

Existing prospective studies have provided valuable insights into long-term treatment outcomes, but they are often limited by sample size and geographic scope. Given resource constraints, recent efforts have focused on standardized datasets to ensure their comparability. In this regard, our core protocol aligns published repository guidelines and incorporates cost-effective diagnostic and follow-up strategies (physical examination, ECG, and echocardiography), making it adaptable to low-resource settings [42,43]. This harmonization will allow for comparisons with other large-scale cohorts worldwide such as the SamiTrop cohort, the NEPA-CHA or the Oxente Chagas Bahia project, contributing to a global understanding of CD [44,45].

One challenge of this multicenter study is the potential variability in laboratorial diagnostic methods, some second-line follow-up assessments, and interpretation of clinical findings across different sites. Additionally, the long follow-up required to assess disease progression and treatment effects poses logistical challenges, including patient retention and resource allocation. However, the benefits of a collaborative research network outweigh these limitations. As part of a collaborative research network, this approach enables us to: a) optimize resources and reduce costs in a context were securing competitive funding for neglected tropical diseases (NTDs) is increasingly challenging; b) increase the potential to generate innovative and high-impact results; c) fosters a supportive learning environment for researchers; and d) strengthen the national research network. From a public health perspective, this initiative can also yield valuable data on the economic burden and the use of health resources. Such data can inform evidence-based public health policies to improve CD detection, treatment, and monitoring, and provide reliable data to guide specific clinical policies and management guidelines for the disease in non-endemic regions.

This prospective, multicenter cohort study has been conceived to generate significant evidence regarding the epidemiology and clinical evolution of CD in a non-endemic setting, evaluate treatment impact and stablish a robust research framework. By addressing key global research questions, such as disease progression, cure biomarkers, and broader aspects of this neglected condition, it will contribute to improving CD management worldwide.

## Author contributions

**Conceptualization:** Pau Bosch-Nicolau, Juan María Herrero-Martínez, Marta Arsuaga, Sandra Chamorro-Tojeiro, Irene Carrillo, Clara Crespillo-Andújar, Aroa Silgado, Ana Pérez-Ayala, Fernando de la Calle-Prieto, José A. Pérez-Molina, Israel Molina.

**Funding acquisition:** Israel Molina.

**Methodology:** Pau Bosch-Nicolau, Juan María Herrero-Martínez, Marta Arsuaga, Sandra Chamorro-Tojeiro, Irene Carrillo, Clara Crespillo-Andújar, Aroa Silgado, Ana Pérez-Ayala, Fernando de la Calle-Prieto, Israel Molina.

**Project administration:** Pau Bosch-Nicolau.

**Supervision:** Pau Bosch-Nicolau, José A. Pérez-Molina, Israel Molina.

**Validation:** Carlos Bea-Serrano, Jara Llenas-García, Sandra Pérez-Recio, Elia Fernández-Pedregal, Laura Prieto-Pérez, José A. Pérez-Molina, Israel Molina.

**Writing – original draft:** Pau Bosch-Nicolau, Juan María Herrero-Martínez, Marta Arsuaga, Sandra Chamorro-Tojeiro, Irene Carrillo, Clara Crespillo-Andújar, Aroa Silgado, Ana Pérez-Ayala, Fernando de la Calle-Prieto.

**Writing – review & editing:** Carlos Bea-Serrano, Jara Llenas-García, Sandra Pérez-Recio, Elia Fernández-Pedregal, Laura Prieto-Pérez, José A. Pérez-Molina, Israel Molina.

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
