## [Decision Letter · Decision Letter 0]

8 Jan 2025

PONE-D-24-53106The Chagas non-endemic cohort (“ChaNoE”): aims and study protocolPLOS ONE

Dear Dr. Bosch-Nicolau,

Thank you for submitting your manuscript to PLOS ONE. After careful consideration, we feel that it has merit but does not fully meet PLOS ONE’s publication criteria as it currently stands. Therefore, we invite you to submit a revised version of the manuscript that addresses the points raised during the review process.

We look forward to receiving your revised manuscript.

Kind regards,

Faham Khamesipour, Ph.D.

Academic Editor

PLOS ONE

Journal Requirements:

“This project was funded by “Proyectos de investigación en salud” program of the Instituto Carlos III, Ministry of Science, Innovation and Universities of the Spanish Government. Grant number: PI22/01894.“

“This project was funded by “Proyectos de investigación en salud” program of the Instituto Carlos III, Ministry of Science, Innovation and Universities of the Spanish Government. Grant number: PI22/01894.”

“This project was funded by “Proyectos de investigación en salud” program of the Instituto Carlos III, Ministry of Science, Innovation and Universities of the Spanish Government. Grant number: PI22/01894.“

**Additional Editor Comments:**

For your submission "The Chagas non-endemic cohort (“ChaNoE”): aims and study protocol" to PLOS ONE to proceed further in the review process, you will need to make major revisions to your manuscript. Revise it as suggested by the reviewer.

Reviewers' comments:

Reviewer's Responses to Questions

**Comments to the Author**

1. Does the manuscript provide a valid rationale for the proposed study, with clearly identified and justified research questions?

Reviewer #1: Yes

Reviewer #2: Yes

Reviewer #3: Yes

2. Is the protocol technically sound and planned in a manner that will lead to a meaningful outcome and allow testing the stated hypotheses?

Reviewer #1: Yes

Reviewer #2: No

Reviewer #3: Yes

3. Is the methodology feasible and described in sufficient detail to allow the work to be replicable?

Reviewer #1: Yes

Reviewer #2: No

Reviewer #3: Yes

4. Have the authors described where all data underlying the findings will be made available when the study is complete?

Reviewer #1: No

Reviewer #2: No

Reviewer #3: Yes

5. Is the manuscript presented in an intelligible fashion and written in standard English?

Reviewer #1: Yes

Reviewer #2: Yes

Reviewer #3: Yes

6. Review Comments to the Author

You may also provide optional suggestions and comments to authors that they might find helpful in planning their study.

Reviewer #1: The authors propose a multi-center study cohort to investigate Chagas disease in a non-endemic country. This is a very important initiative and I congratulate the authors.

Comments:

1) Please, include in the objectives the study of clinical end-points. Hard end-points, such as all-cause mortality and heart transplant, but also secondary end-points, such as, admission due to heart failure, cardiac device implantation, and stroke. As a lot of data will be collected, it is very important to understand how these variables are associated with clinical outcomes.

2) Please, clarify criteria for disease progression. Do you mean only progression from indeterminate to cardiac form (A to B/C or D) ? Or also progression within stages of the cardiac form (B to C or D)? Will also study progression from indeterminate to digestive form?

3) Please, clarify the criteria for cure that will be followed.

4) Please, clarify if a patient will be classified as B if changes are present in the echocardiogram even in the presence of a normal ECG. I ask this because of the following sentence : “CCC will be defined by the presence of segmental or global left ventricle (LV) or right ventricle (RV) systolic function impairment, increased end-diastolic diameter of the LV (>55mm), ventricular aneurysms or cardiac thrombus”. Usually, an abnormal ECG is always necessary for a patient to be diagnosed with CCC.

5) Table 2. A 24-Holter is always necessary in the initial evaluation of all patients with CCC. Specially if the study objective is the evaluation of prognosis1.

6) Please, evaluate the inclusion of biochemistry and blood count in the initial evaluation and follow-up. A poor kidney function, anemia, low sodium are all associated with a dismal outcome in patients with heart failure.

7) Table 2. Echocardiography in stage A (indeterminate) can be at every 5 years. B1 patients should undergo echocardiograms more frequently as they already have changes in ECG: every two years, and B2 and C patients every year1.

8) Table 6. Correct fifth line “5mg/day (bid or tid)”

9) Benznidazole treatment dose recommendation is different in Brazil. “For adults with chronic CD, benznidazole is administered orally at the dosage of 5mg/kg/day divided into two or three doses, for 60 days, with a recommended maximum dosage of 300mg/day. For individuals with acute CD, the dose can be of as much as 10mg/kg/day. For individuals weighing over 60 kg, the therapeutic schedule can be extended to achieve the ideal target dose, maintaining 300 mg as the daily limit to prevent adverse events. The 300mg benznidazole regimen can be used for the number of days equivalent to the individual’s weight, limited to a total of 80 days even for individuals weighing over 80 kg”2. The authors’ table 6 suggests that there is no upper limit for benznidazole daily dose. For example, a patient with 90 kg would be prescribed 450 mg/day but this can increase the risk of adverse effects.

10) Include in all Tables the references the authors used.

11) Statistical analysis. Include how prognosis will be evaluated. Survival curves, logistic regression, non-adjusted or adjusted, ROC curves? Please, clarify

12) Dissemination plan. Is there a plan to make data available in public repositories?

13) Discussion. There are other papers describing guides for the generation of databases or repositories for Chagas disease cohorts. I invite the authors to read those papers3,4.

1 - Saraiva RM, Mediano MFF, Mendes FS, Sperandio da Silva GM, Veloso HH, et al. Chagas heart disease: An overview of diagnosis, manifestations, treatment, and care. World J Cardiol. 2021 Dec 26;13(12):654-675.

2 - Marin-Neto JA, Rassi A Jr, Oliveira GMM, Correia LCL, Ramos Júnior AN, et al. SBC Guideline on the Diagnosis and Treatment of Patients with Cardiomyopathy of Chagas Disease - 2023. Arq Bras Cardiol. 2023 Jun 26;120(6):e20230269.

3-González Martínez A, Losada-Galván I, Gabaldón-Figueira JC, Martínez-Peinado N, Saraiva RM, et al. A standardized clinical database for research in Chagas disease: The NHEPACHA network. PLoS Negl Trop Dis. 2024 Aug 15;18(8):e0012364.

4-Martínez-Peinado N, Gabaldón-Figueira JC, Rodrigues Ferreira R, Carmen Thomas M, López MC, et al. A guide for the generation of repositories of clinical samples for research on Chagas disease. PLoS Negl Trop Dis. 2024 Aug 15;18(8):e0012166.

Reviewer #2: Summary:

This article describes the literature behind, and some of what the Spanish Chagas non-endemic cohort (ChaNoE), a multi-center initiative based in Spain, seeks to accomplish. Their aim is to improve the understanding and management of Chagas disease (CD) in non-endemic regions, and to contribute evidence regarding clinical progression, biomarkers, and overall understanding of Chagas disease. The study focuses on:

They aim to recruit individuals with chronic CD, standardize diagnostic and treatment protocols, monitor disease progression, and create a biobank to facilitate biomarker research for progression and treatment efficacy. A serum biobank is being established to identify and validate biomarkers for disease progression and treatment response. This is a critical need in CD research. The Spanish experience is useful for cohorts elsewhere in non-endemic areas. The authors do a good job reviewing existing literature on CD, including its classification and management. However, the article reads more like a review of the literature rather than a research study protocol. There are many details missing from this description, such as inclusion and exclusion criteria, detailed timelines, where the samples will be taken and stored, which specific serology tests will be used to diagnose CD, who will recruit participants, etc. This is a great idea for a project, and the achievement of the cohort will be very beneficial. However, in order to publish the protocol, it needs to provide much more detail.

Introduction:

• In line 92, it says that treatment is recommended for people in the chronic phase. This is not exactly accurate. The recommendations depend on the age of the individual and their circumstances, as the BENEFIT trial showed that treatment in older individuals with advanced cardiomyopathy did not result in significant improvement. Treatment is more strongly recommended for younger patients. However, guidelines vary by region. Please be specific about the treatment indications for chronic disease in Spain.

• The introduction should typically end with the overall aim of the study/work. In this case, it could be more specific about the particular context in Spain and state that the article will describe the cohort and its intent.

Methods:

• Overall, a large proportion of the paragraphs in the methods section seem more like introductory paragraphs, describing the literature. Please remember that the introduction or background section of an article reviews existing literature, and this often happens again in the discussion section when comparing findings to existing studies. Please reorganize the article to clarify each section. The methods should describe the project itself rather than the literature.

• Line 179: Where is the annual incidence of Chagas cardiomyopathy sourced from? There is a wide range in the literature, so please include more references.

• Line 181: Diagnosis and categorization of CCC also vary. Many newer articles include NYHA severity classes in the definitions, so please adjust to clarify this – you already touch on this in line 195.

• Line 253: Digestive tract involvement is less than 15%. Please correct. As you mentioned, some places in the Southern Cone have a higher proportion of digestive tract involvement, but this is not a generalization for Chagas. The Brazilian article identifies female sex as a predictor; however, this line could mislead readers. It is well known that only some genetic variants of T. cruzi are associated with digestive forms—this is the biggest association, not female sex.

• The methods section reads more like a review of the literature than a description of what the ChaNoE cohort will entail. It is important to be very detailed about your specific cohort, rather than existing literature. I see a review of the current recommendations for Chagas, but it’s not clear how the timeline works, who the team will consist of, how patients will be followed up, and how referrals will be made within the specific hospitals. Are these patients insured? Will they be using free national insurance? How will the team ensure that all follow-up care is completed? Will different sites use either benznidazole or nifurtimox for treatment, and how will that be decided? Based on the broad description, it sounds like these participants will be treated for all possible complications of Chagas disease.

• We don’t know the specific inclusion or exclusion criteria for participants, such as age ranges and eligibility. This is only described briefly in the abstract and not in detail.

• Line 331 says that patients will have easy access to their healthcare provider; however, it is unclear if this provider is part of the study or if it refers to a regular visit, with medical charts being reviewed.

Discussion:

• The discussion reads somewhat broadly about Chagas disease, rather than specifically discussing the cohort study in the context of previous work.

Reviewer #3: Dear Editor and Authors,

Thank you for the opportunity to review the manuscript titled "The Chagas Non-Endemic Cohort (“ChaNoE”): Aims and Study Protocol." This is an intriguing and valuable project. The manuscript effectively describes the establishment of a Spanish cohort designed to identify and monitor chronic cases of Chagas disease. The study employs a modern clinical-laboratory approach and clearly articulates its objectives and rationale in a well-justified manner. The manuscript addresses an important public health challenge: Chagas disease in non-endemic regions, a topic that remains underrepresented in current research. The focus on Spain—a country with a significant population of individuals living with Chagas disease due to migration—highlights the study’s relevance and potential impact. However, before the manuscript is accepted, several questions and concerns need to be addressed, as outlined below:

Major comments

1. The authors mention multiple times that the data obtained from the “ChaNoE” cohort will be compared with other cohorts worldwide. However, the manuscript does not specify these cohorts. Including examples of similar studies, such as Oxente Chagas Bahia, Samitrop, TESEU, and others, would enhance the manuscript by informing readers of relevant ongoing research efforts globally and providing context for the comparative analyses planned in this study.

2. While the protocol is thorough, some methodological aspects, such as the specific strategies for data standardization across centers, are underexplained. A clearer description of quality control for data and biobank management is needed.

3. The study’s reliance on multiple centers for participant recruitment could face logistical hurdles, especially in ensuring consistent diagnostic criteria and follow-up. Comment!

4. Although PCR is mentioned, its limited sensitivity in chronic cases raises concerns. The manuscript could better address how it will account for false negatives or inconsistent results. Explain how the study will mitigate the limitations of current diagnostic tools, such as PCR, and whether alternative or complementary methods will be employed.

5. While the protocol outlines periodic evaluations, the manuscript does not provide sufficient detail on how adherence to follow-up will be ensured, particularly for participants in non-urban areas. Discuss strategies to overcome potential challenges in participant recruitment and retention, including communication methods and incentives.

6. Inclusion criteria requiring diagnosis by two serological tests might exclude some cases, potentially limiting generalizability.

7. Add visual elements, such as flowcharts or timelines, to illustrate the study design, recruitment process, and follow-up schedule.

8. The potential financial burden of implementing such protocols in other settings, including low-resource regions, is not addressed. Include a discussion of how the findings and methodologies could be adapted for low-resource or other non-endemic settings. In addition, provide examples of how the study outcomes will directly influence public health policies or clinical practices in non-endemic areas.

9. Include plans for biobank management and data sharing policies.

Minor comments

1. Line 44: Italicize Trypanosoma cruzi.

2. Line 49: Clarify the term "heterogeneous." As it stands, the meaning is unclear and could benefit from further explanation or rephrasing?

3. Line 81: Remove the word "also." The phrase, as written, might imply that vectorial transmission occurs in non-endemic areas, which could lead to misinterpretation.

4. Line 403: Specify the microbiological tests used for diagnosing Chagas disease. Wouldn't these be more accurately described as parasitological tests?

5. Line 470: Capitalize "Chagas."

7. PLOS authors have the option to publish the peer review history of their article (what does this mean? ). If published, this will include your full peer review and any attached files.

**Do you want your identity to be public for this peer review?** For information about this choice, including consent withdrawal, please see our Privacy Policy .

Reviewer #1: No

Reviewer #2: No

Reviewer #3: **Yes: ** Fred Luciano Neves Santos

---

## [Author Response · Author response to Decision Letter 1]

15 Feb 2025

Response to Reviewers:

Reviewer #1: The authors propose a multi-center study cohort to investigate Chagas disease in a non-endemic country. This is a very important initiative and I congratulate the authors.

Comments:

1) Please, include in the objectives the study of clinical end-points. Hard end-points, such as all-cause mortality and heart transplant, but also secondary end-points, such as, admission due to heart failure, cardiac device implantation, and stroke. As a lot of data will be collected, it is very important to understand how these variables are associated with clinical outcomes.

Thank you for your valuable comment. Upon reviewing our manuscript, we realized that we had omitted a detailed description of the clinical and laboratory outcomes, as well as the criteria for their measurement and definition. Therefore, we have now included both primary and secondary outcomes in the "Aims and Objectives" section. In accordance with the reviewer’s suggestion, we have specified the criteria for disease progression in both the cardiac and digestive forms and clarified the definition of cure.

2) Please, clarify criteria for disease progression. Do you mean only progression from indeterminate to cardiac form (A to B/C or D) ? Or also progression within stages of the cardiac form (B to C or D)? Will also study progression from indeterminate to digestive form?

Please, refer to our response to your previous comment.

3) Please, clarify the criteria for cure that will be followed.

Please, refer to our response to your previous comment.

4) Please, clarify if a patient will be classified as B if changes are present in the echocardiogram even in the presence of a normal ECG. I ask this because of the following sentence: “CCC will be defined by the presence of segmental or global left ventricle (LV) or right ventricle (RV) systolic function impairment, increased end-diastolic diameter of the LV (>55mm), ventricular aneurysms or cardiac thrombus”. Usually, an abnormal ECG is always necessary for a patient to be diagnosed with CCC.

Although this scenario is infrequent, different authors have proposed that echocardiography can also detect early alterations in patients with normal ECG findings*. While this approach may not be feasible in all settings, we believe that such alterations should be considered, as these patients are likely at higher risk of further progression compared to those with completely normal findings. Further evidence is needed in this regard, and we hope that the present cohort will contribute valuable data. We acknowledge that this may pose challenges in comparing our results with other cohorts; however, we plan to conduct subgroup analyses for these patients.

To ensure clarity, we have modified the relevant sentence to: “CCC will be defined by the presence of segmental or global left ventricle (LV) or right ventricle (RV) systolic function impairment, increased end-diastolic diameter of the LV (>55mm), ventricular aneurysms, or cardiac thrombus, even in the presence of a normal ECG.” Additionally, we have introduced the quotation “2” to “Table 2 Classification and follow-up of…”

* Laynez-Roldán P, Losada-Galván I, Posada E, Ávila L de la T, Casellas A, Sanz S, et al. Characterization of Latin American migrants at risk for Trypanosoma cruzi infection in a non-endemic setting. Insights into initial evaluation of cardiac and digestive involvement. PLoS Negl Trop Dis 2023;17. https://doi.org/10.1371/JOURNAL.PNTD.0011330.

* Ralston K, Zaidel E, Acquatella H, Barbosa MM, Narula J, Nakagama Y, et al. WHF Recommendations for the Use of Echocardiography in Chagas Disease. Glob Heart 2023;18. https://doi.org/10.5334/GH.1207.

5) Table 2. A 24-Holter is always necessary in the initial evaluation of all patients with CCC. Specially if the study objective is the evaluation of prognosis1.

We anticipated this comment, as it has been a point of discussion among infectious disease specialists, cardiologists, and professionals working in less complex institutions. While we acknowledge the limitation of not performing a baseline Holter in all patients, we ultimately decided to conduct Holter monitoring in all patients classified as CCC (from stage A to D), as well as in those with normal ECG but abnormal echocardiographic findings or suggestive symptoms, as specified in Table 2. This decision was made after considering the current lack of strong evidence supporting the cost-effectiveness of a 24-hour Holter in asymptomatic patients with normal ECG and echocardiographic findings.

6) Please, evaluate the inclusion of biochemistry and blood count in the initial evaluation and follow-up. A poor kidney function, anemia, low sodium are all associated with a dismal outcome in patients with heart failure.

We fully agree with the reviewer that these parameters are essential at inclusion. Consequently, we have incorporated biochemical and hematological assessments into the initial evaluation. In follow-up, we decided to maintain it optional depending on each case (many of our patients are young individuals with no comorbidities and sometimes they already have other blood tests performed by their family doctor.

7) Table 2. Echocardiography in stage A (indeterminate) can be at every 5 years. B1 patients should undergo echocardiograms more frequently as they already have changes in ECG: every two years, and B2 and C patients every year1.

Thank you for your suggestion. We have adjusted the follow-up intervals for B1 patients accordingly. The follow-up periods for B2 and C patients were already in line with the recommendation.

8) Table 6. Correct fifth line “5mg/day (bid or tid)”

The correction has been made.

9) Benznidazole treatment dose recommendation is different in Brazil. “For adults with chronic CD, benznidazole is administered orally at the dosage of 5mg/kg/day divided into two or three doses, for 60 days, with a recommended maximum dosage of 300mg/day. For individuals with acute CD, the dose can be of as much as 10mg/kg/day. For individuals weighing over 60 kg, the therapeutic schedule can be extended to achieve the ideal target dose, maintaining 300 mg as the daily limit to prevent adverse events. The 300mg benznidazole regimen can be used for the number of days equivalent to the individual’s weight, limited to a total of 80 days even for individuals weighing over 80 kg”2. The author’s table 6 suggests that there is no upper limit for benznidazole daily dose. For example, a patient with 90 kg would be prescribed 450 mg/day but this can increase the risk of adverse effects.

In Spain, we have traditionally used weight-adjusted dosing, extending treatment only in specific cases. Following discussions with participating centers, we found that while a few centers have used up to 400mg/day, most adhere to the 300mg/day limit when weight does not exceed 100kg. To clarify this, we have added a note after Table 6.

Regarding adverse events, the literature presents conflicting data on the relationship between dosage and toxicity. Notably, a recently published clinical trial (The Multibenz Trial, PMID: 38218195) employed a 400mg/day dosage without observing a significant increase in adverse events.

10) Include in all Tables the references the authors used.

We have now included references in all tables except Table 7, which is based on our own consensus derived from the available evidence.

11) Statistical analysis. Include how prognosis will be evaluated. Survival curves, logistic regression, non-adjusted or adjusted, ROC curves? Please, clarify

To address this, we have added a paragraph to the "Data Management and Statistical Analysis" section detailing how we will analyze survival outcomes and potential confounders.

12) Dissemination plan. Is there a plan to make data available in public repositories?

As approved by the Ethics Committee, data will be shared with appropriate academic parties upon request to the chief investigator. We have added a paragraph clarifying this in the "Ethical Considerations and Dissemination Plans" section.

13) Discussion. There are other papers describing guides for the generation of databases or repositories for Chagas disease cohorts. I invite the authors to read those papers3,4.

1 - Saraiva RM, Mediano MFF, Mendes FS, Sperandio da Silva GM, Veloso HH, et al. Chagas heart disease: An overview of diagnosis, manifestations, treatment, and care. World J Cardiol. 2021 Dec 26;13(12):654-675.

2 - Marin-Neto JA, Rassi A Jr, Oliveira GMM, Correia LCL, Ramos Júnior AN, et al. SBC Guideline on the Diagnosis and Treatment of Patients with Cardiomyopathy of Chagas Disease - 2023. Arq Bras Cardiol. 2023 Jun 26;120(6):e20230269.

3-González Martínez A, Losada-Galván I, Gabaldón-Figueira JC, Martínez-Peinado N, Saraiva RM, et al. A standardized clinical database for research in Chagas disease: The NHEPACHA network. PLoS Negl Trop Dis. 2024 Aug 15;18(8):e0012364.

4-Martínez-Peinado N, Gabaldón-Figueira JC, Rodrigues Ferreira R, Carmen Thomas M, López MC, et al. A guide for the generation of repositories of clinical samples for research on Chagas disease. PLoS Negl Trop Dis. 2024 Aug 15;18(8):e0012166.

Thank you for this suggestion. Some of our investigators have already participated in developing consensus guidelines to harmonize and facilitate comparability in prospective Chagas disease studies. Recognizing the importance of this issue, we have incorporated the suggested references and added relevant sentences to the third paragraph of the Discussion.

Reviewer #2: Summary:

This article describes the literature behind, and some of what the Spanish Chagas non-endemic cohort (ChaNoE), a multi-center initiative based in Spain, seeks to accomplish. Their aim is to improve the understanding and management of Chagas disease (CD) in non-endemic regions, and to contribute evidence regarding clinical progression, biomarkers, and overall understanding of Chagas disease. The study focuses on: They aim to recruit individuals with chronic CD, standardize diagnostic and treatment protocols, monitor disease progression, and create a biobank to facilitate biomarker research for progression and treatment efficacy. A serum biobank is being established to identify and validate biomarkers for disease progression and treatment response. This is a critical need in CD research. The Spanish experience is useful for cohorts elsewhere in non-endemic areas. The authors do a good job reviewing existing literature on CD, including its classification and management. However, the article reads more like a review of literature rather than a research study protocol. There are many details missing from this description, such as inclusion and exclusion criteria, detailed timelines, where the samples will be taken and stored, which specific serology tests will be used to diagnose CD, who will recruit participants, etc. This is a great idea for a project, and the achievement of the cohort will be very beneficial. However, in order to publish the protocol, it needs to provide much more detail.

Thank you for all your comments. We really appreciate them and think that the following changes will improve the article’s quality. We will answer the specific questions below.

Introduction:

• In line 92, it says that treatment is recommended for people in the chronic phase. This is not exactly accurate. The recommendations depend on the age of the individual and their circumstances, as the BENEFIT trial showed that treatment in older individuals with advanced cardiomyopathy did not result in significant improvement. Treatment is more strongly recommended for younger patients. However, guidelines vary by region. Please be specific about the treatment indications for chronic disease in Spain.

Thank you for your comments. As you mentioned, there is an ongoing debate on treatment benefit and indications. There are several controversies and evidence in favor and against treating chronic patients, especially those with advanced age and mild cardiopathy. The BENEFIT trial showed us that in moderate or advanced cardiomyopathy, parasitological treatment offers no effect on disease progression, but in other groups we consider the evidence provided by this study as limited. However, this evidence has been widely debated, particularly regarding its applicability to all chronic patients. During last years, many countries have changed their protocols in favor of being laxer in its indication, and for example, Brazil has started to treat much more patients during last years.

Referring to age limits, we consider that the habitual 50-year-age limit has been based on limited life expectancy in most endemic regions. However, these populations, when migrating, have showed a tendency to equal the native population which in Spain the mean is around 83 years old. In Spain we already have much experience treating elder patients and we are close to publish our results supporting this indication during the present year. In summary, considering the available evidence and our own experience, we have set our indications in Table 7.

• The introduction should typically end with the overall aim of the study/work. In this case, it could be more specific about the particular context in Spain and state that the article will describe the cohort and its intent.

We appreciate your suggestion. We have revised the final paragraph of the introduction to explicitly state the study’s objectives within the Spanish context, ensuring alignment with standard scientific structuring.

Methods:

• Overall, a large proportion of the paragraphs in the methods section seem more like introductory paragraphs, describing the literature. Please remember that the introduction or background section of an article reviews existing literature, and this often happens again in the discussion section when comparing findings to existing studies. Please reorganize the article to clarify each section. The methods should describe the project itself rather than the literature.

Thank you for this comment. We completely agree with you that the previous structure seems more like a literature review with an embedded protocol. In this sense, we have moved some of the information to the discussion, maintaining only those paragraphs that are essential to understand the protocol and replicate it in the methods section. With your recommendation now the protocol is easier to follow.

• Line 179: Where is the annual incidence of Chagas cardiomyopathy sourced from? There is a wide range in the literature, so please include more references.

We have added two new references, including a systematic review and meta-analysis. This ensures a more comprehensive and evidence-based estimate of the disease burden.

• Line 181: Diagnosis and categorization of CCC also vary. Many newer articles include NYHA severity classes in the definitions, so please adjust to clarify this – you already touch on this in line 195.

We acknowledge the importance of incorporating NYHA classification into our study. As a result, we have ensured that NYHA classification is clearly specified in Table 2, both for baseline assessment and follow-up evaluations, to facilitate a more standardized and clinically relevant classification of disease severity.

• Line 253: Digestive tract involvement is less than 15%. Please correct. As you mentioned, some places in the Southern Cone have a higher proportion of digestive tract involvement, but this is not a generalization for Chagas. The Brazilian article identifies female sex as a predictor; however, this line could mislead readers. It is well known that only some genetic variants of T. cruzi are associated with digestive forms—this is the biggest association, not female sex.

We have clarified that the figures of prevalence of digestive involvement refers to our setting, while also noting that rates are higher in certain regions of the Southern Cone. Additionally, we have streamlined the discussion on risk factors for digestive forms to maintain focus on the study’s objectives.

• The methods section reads more like a review of the literature than a description of what the ChaNoE cohort will entail. It is important to be very detailed about your specific cohort, rather than existing literature. I see a review of the current recommendations

---

## [Decision Letter · Decision Letter 1]

24 Feb 2025

The Chagas non-endemic (ChaNoE) cohort: aims and study protocol

PONE-D-24-53106R1

Dear Dr. Bosch-Nicolau,

We’re pleased to inform you that your manuscript has been judged scientifically suitable for publication and will be formally accepted for publication once it meets all outstanding technical requirements.

Kind regards,

Faham Khamesipour, Ph.D.

Academic Editor

PLOS ONE

Additional Editor Comments (optional):

I recommend the manuscript for publication without further revisions.

Reviewers' comments:

Reviewer's Responses to Questions

**Comments to the Author**

1. Does the manuscript provide a valid rationale for the proposed study, with clearly identified and justified research questions?

Reviewer #1: Yes

Reviewer #3: Yes

2. Is the protocol technically sound and planned in a manner that will lead to a meaningful outcome and allow testing the stated hypotheses?

Reviewer #1: Yes

Reviewer #3: Yes

3. Is the methodology feasible and described in sufficient detail to allow the work to be replicable?

Reviewer #1: Yes

Reviewer #3: Yes

4. Have the authors described where all data underlying the findings will be made available when the study is complete?

Reviewer #1: Yes

Reviewer #3: Yes

5. Is the manuscript presented in an intelligible fashion and written in standard English?

Reviewer #1: Yes

Reviewer #3: Yes

6. Review Comments to the Author

You may also provide optional suggestions and comments to authors that they might find helpful in planning their study.

Reviewer #1: All questions raised by the reviewer were properly addressed by the authors. No further comments.

Reviewer #3: The authors have incorporated all my suggestions into the manuscript. Following the modifications suggested by all reviewers, the project “ChaNoE” is now clearly presented. Given these strengths, I recommend the manuscript for publication in PLOS ONE without further revisions.

7. PLOS authors have the option to publish the peer review history of their article (what does this mean? ). If published, this will include your full peer review and any attached files.

**Do you want your identity to be public for this peer review?** For information about this choice, including consent withdrawal, please see our Privacy Policy .

Reviewer #1: **Yes: ** Roberto Saraiva

Reviewer #3: **Yes: ** FRED LUCIANO NEVES SANTOS

---

## [Editor Report · Acceptance letter]

PONE-D-24-53106R1

PLOS ONE

Dear Dr. Bosch-Nicolau,

I'm pleased to inform you that your manuscript has been deemed suitable for publication in PLOS ONE. Congratulations! Your manuscript is now being handed over to our production team.

Kind regards,

on behalf of

Dr. Faham Khamesipour

Academic Editor

PLOS ONE